# The Flourishing Camel Milk Market and Concerns about Animal Welfare and Legislation

**DOI:** 10.3390/ani13010047

**Published:** 2022-12-22

**Authors:** Marcel Smits, Han Joosten, Bernard Faye, Pamela A. Burger

**Affiliations:** 1European Camel Research Society, Johanniterlaan 7, 6721 XX Bennekom, The Netherlands; 2Emeritus Professor Microbiology, Chemin de Crocus 1, 1073 Mollie Margot, Switzerland; 3UMR SELMET, CIRAD-ES, Campus International de Baillarguet, 34398 Montpellier, France; 4Research Institute of Wildlife Ecology, University of Veterinary Medicine, Savoyenstrasse 1, 1160 Vienna, Austria

**Keywords:** dromedary, domestication, Old World camels, sustainable animal breeding, genetic diversity, inbreeding, hobby animal

## Abstract

**Simple Summary:**

Until the beginning of this century, dromedaries were mainly utilized as multi-purpose animals, suitable for various activities like transport and production of meat, milk, and wool. In recent decades, however, the production of dromedary milk has increased constantly, not only as staple food in marginal eco-agricultural desert regions of the Global South but also in the Global North, due to presumed health benefits. The enlarged number of dromedaries kept in dromedary dairies has changed the susceptibility of these dromedaries to diseases. Nutrition and social behaviour have also changed as a result. In addition to these influences on animal welfare, the gene composition changes. Protocols for checking animal safety monitor overall animal welfare. Gene banks are going to prevent inbreeding and unwanted gene change. Governments are working on improved regulation concerning the food safety of dromedary milk and on drawing up legislation to ensure the well-being of dromedaries. However, this legislation is still in a preliminary phase requiring sound scientific support to identify and correct illegalities and other imperfections well in advance.

**Abstract:**

The worldwide dromedary milk production has increased sharply since the beginning of this century due to prolonged shelf life, improved food-safety and perceived health benefits. Scientific confirmation of health claims will expand the market of dromedary milk further. As a result, more and more dromedaries will be bred for one purpose only: the highest possible milk production. However, intensive dromedary farming systems have consequences for animal welfare and may lead to genetic changes. Tighter regulations will be implemented to restrict commercialization of raw milk. Protocols controlling welfare of dromedaries and gene databases of milk-dromedaries will prevent negative consequences of intensive farming. In countries where dromedaries have only recently been introduced as production animal, legislators have limited expertise on this species. This is exemplified by an assessment on behalf of the Dutch government, recommending prohibiting keeping this species from 2024 onwards because the dromedary was deemed to be insufficiently domesticated. Implementation of this recommendation in Dutch law would have devastating effects on existing dromedary farms and could also pave the way for adopting similar measures in other European countries. In this paper it is shown that the Dutch assessment lacks scientific rigor. Awareness of breeders and legislators for the increasing knowledge about dromedaries and their products would strengthen the position of dromedaries as one of the most adapted and sustainable animals.

## 1. Introduction

Old World camels, the one-humped *Camelus dromedarius* and the two-humped *Camelus bactrianus* have served humans in cross-continental caravans, transporting people and goods, connecting different cultures of Arabia, the Near east, and North Africa, and providing milk, meat, wool and draft power since their domestication around 3000–6000 years ago [1,2]. In addition to these domesticated types, a third camel species is recognized today that has never been domesticated and is the only remaining wild representative of the genus *Camelus*, the two-humped *Camelus ferus* (Przewalski, 1874) [3,4]. It is “critically endangered” (IUCN 2018) and exists only in the Mongolian Great Gobi and in the Chinese Lop Nur and Taklamakan deserts. Nowadays, the domesticated dromedary and Bactrian camels are considered as sustainable, multipurpose livestock species [5]. For a long time, fresh camel milk was consumed only by nomads and freely given as a gift for guests. Consequently, camel milk was not considered merchandise and its sale was often taboo [2]. Beyond their traditional contribution to regional meat markets [6], camels are increasingly kept for today for two specific purposes, (i) for performing sporting achievements, e.g., to participate in camel races or beauty contests, especially in Middle Eastern countries; and (ii) for becoming increasingly an important producer of camel milk [7], not only gaining ground in countries where camel milk has traditionally been consumed, but also in European countries where the number of camel dairies is increasing [8,9].

However, the targeted breeding of single-purpose camels with the aim of more efficient milk production leads to genetic change of the animals. Despite the contemporary development and changing ways of camel breeding practices and management, little attention is paid to the consequences for the welfare of the animals. In recent years, however, scientific knowledge has increased [10,11] and legislation is being developed in various countries with the aim of regulating animal welfare. To protect animal welfare, “positive” domestic and hobby animals’ lists are adopted. The intention of these lists is that only the possession and sale of animal species, that do not pose a substantial risk to human and animal well-being is allowed [12].

In this Commentary we identify factors that influence the increased milk production that contributes to further development of the single-purpose milk dromedary and factors that influence the dromedary welfare. Furthermore, we describe legislative attempts to protect the food safety of dromedary milk and animal welfare. We comment on the decision to exclude the dromedary from the “positive” list of domestic and hobby animals in the Netherlands. Finally, we summarize evidence of dromedary biology and domestication to obviate misconceptions of scientific literature, which might have severe consequences for the young but thriving camel milk market in Europe and beyond.

## 2. The Increase of The Camel Milk Market

During the time when dromedary milk was consumed only by nomads, the raw product did not undergo any transformation, except fermentation to extend shelf life in arid and hot regions, where sufficient cooling of milk was not possible [13]. At the beginning of the 21st century, dromedary milk entered the national and international markets because freezing of the milk turned out to be possible without changing taste and composition [14]. As a result, the shelf life of dromedary milk could be significantly extended to up to at least one year after production date. More consumers could be reached since transportation was no longer limited. The market expanded even further when mechanical milking was introduced. Meanwhile, several kinds of dromedary milk products have been developed such as milk powder, cheese, chocolate, ice cream, baby food and beauty products [8,9]. According to FAO statistics, the global production of dromedary milk experienced more than 8% annual growth in the period 2009–2019 [15,16].

Although most of the dromedary milk is still sold directly to consumers via retailers, it can be expected that the shares of online sales via internet and e-commerce will gain importance [17]. The FAO forecasts that over the next 5 years the dromedary milk market will continue to grow by an average of 8% per annum [16,18]. Over the past decade, consumers’ interest in dromedary milk has improved particularly in Europe and the United States, thanks to a growing number of foreign consumers coming from countries where dromedary milk is part of their tradition [16,18].

### 2.1. Strenghts of the Dromedary Milk Market from European Perspectives

#### 2.1.1. Sustainability

In view of increasing climate change, dromedaries (and Bactrian camels) can be considered as one of the most sustainable livestock species. Ammonia emission from dromedary is low, i.e., 10–15% of those from cows [19]. This ammonia emission decreases further, when urine from camels is collected separately, for its perceived health promoting properties [20,21,22]. In addition, due to its feeding behaviour, camel is able to clean rangelands invaded by brambles, thistles and nettles in abandoned areas allowing strong economy of shredding. Thanks to its soft feet, the pressure on agriculture soil is quite lower than for cow (2.6 newton/cm² vs. 8.86 newton/cm², respectively). Importantly, with a water turnover of 80 mL/kg/24 h, the water requirements of the camel is half of the cow’s requirements (160 mL/kg/24 h). Finally, the methane emission by the dromedary is much lower than that of ruminants. The methane emission of Australia’s feral camels corresponds only to 1–2% of that produced by the countries’ domestic ruminants [23].

#### 2.1.2. Adaptation to Climate Change

Dromedaries are known to be excellent at adapting to widely varying climatic conditions. In Africa, the distribution area of dromedary camels is increasing and new “camel’s countries” appeared such as Nigeria, Uganda, Tanzania, Kenya or even Botswana where cattle breeders are switching their bovines to camels [24]. Dromedaries and Bactrian camels also appear to be resistant to the climate of Europe, as evidenced by European milk production, which corresponds to that of Arabian dromedaries, which are milked under domestic desert conditions [25].

#### 2.1.3. Health Promoting Properties

The increasing popularity of dromedary milk is probably also due to its perceived health-promoting properties [26], including the medicinal properties summarized in Table 1. Dromedary milk contains more iron and vitamin C than cow’s milk. In addition, dromedary milk from Europe and the Middle-East contains less cholesterol and fat [27,28]. All these properties together increase the popularity of dromedary milk as a “superfood” [29]. The traditionally assumed health-promoting properties of dromedary milk have not yet been conclusively proven scientifically [26]. Randomized placebo-controlled trials are crucial to confirm the effectiveness of dromedary milk. Only for autism a few of such trials were performed. However, their treatment frames were too short to conclude that dromedary milk helps the treatment of autism. For the treatment of diabetes a few controlled trials with less than 50 diabetic patients were published. The other health promoting properties have only been studied in observational or hypothesis investigating studies (Table 1).

### 2.2. Challenges of a Sustainable Camel Milk Market

#### 2.2.1. Genetic Development

Breeders of dromedaries select for animals with the highest milk production. However, the genetic diversity of these single-purpose animals [2] can be compromised and intensive selection can lead to progressive impoverishment of the gene pool as shown in cattle [40]. To prevent genetic erosion, it is being investigated how to improve camel breeding while maintaining genetic diversity as to preserve the specific physiological adaptation to the desert habitat. In addition, in Europe, due to a lack of camel import possibilities and missing studbooks of dromedaries, there was no effective way to prevent the negative consequences of inbreeding. However, new scientific tools enable capturing exactly those places in the genome (SNPs) that provide information about potential harmful alleles next to inbreeding values [41], as well as the general genetic relationship of the animals. Therefore, a database of European dromedaries is in progress, which can mitigate the progressive impoverishment of the gene pool [42].

#### 2.2.2. Price Comparison

Dromedary milk is relatively expensive compared to cow’s milk (Figure 1). This high price is caused by the fact that a dromedary produces only up to 6–7 L of milk per day, whereas a cow can provide between 30 and 40 L. In addition, dromedaries give milk for a much shorter period of time as the lactation period is 10–12 months. During the pregnancy of 13 months dromedaries do not produce milk, unlike cows. Only when the calf has pre-drunk and then remains in the vicinity of the mother, the milking machine can be successfully set in motion. When the newborn calf is not accepted by the mother or when the calf dies, the mother usually will not produce milk anymore. In a period of 2.5 years, a dromedary usually delivers milk for no more than 10–12 months. The average milk production per cow has doubled over the last 50 years [43]. It is to be expected that the milk production per dromedary will increase as result of selection and experiences of camel dairy farms [44].

#### 2.2.3. Reproduction

Dromedaries are seasonal breeders with a relatively short breeding season during the cooler months. The onset of the breeding season can be influenced by local environmental factors such as temperature and pasture availability, although decreased libido of the males as the environmental temperature increases, is also a factor. Oestrous behaviour is highly variable in duration and intensity and is, therefore, unreliable for the detection of oestrus and difficult to relate to follicular activity in the ovaries. Camels are induced ovulators and thus normally only ovulate in response to mating [45]. Use of assisted reproduction for the improvement of milk production in dairy dromedaries is investigated at a large camel dairy [46]. The application of artificial insemination in camels is hindered by difficulties involved in collection, as well as handling the semen due to the viscous nature of the seminal plasma [47]. Therefore, artificial insemination of camels is far less successful than that of cows.

#### 2.2.4. Consumption and Selling of Raw Camel Milk

Many consumers of dromedary milk prefer the unpasteurized product for various reasons: raw milk is often claimed to have better organoleptic properties and a heat treatment is believed to be detrimental for its nutritional value and purported health promoting properties. In some communities there is even a cultural taboo on boiling dromedary milk [43,48]. However, raw milk can contain hazardous microorganisms either as a result of direct shedding into the milk or through contamination from the environment during milk collection. It is, therefore, no surprise that raw camel milk also has been reported as vehicle in several disease outbreaks [49] and that it is considered to represent a significant public health risk [50,51,52]. The scientific basis of health claims with respect to raw milk is generally considered to be very weak [53]. Sale of raw milk is, therefore, restricted or even forbidden in many countries [53,54,55].

Several measures can be taken to reduce contamination of milk with pathogenic microorganisms such as improving hygiene during milking and maintaining good animal health, but the impact of such measures is limited [50]. Monitoring of the milk quality by frequent microbiological analysis is not very effective either [56]. Heat treatment, therefore, remains the most reliable and cost effective approach to render milk pathogen free, but alternative non-thermal solutions such as microfiltration and high pressure treatment are available to produce a safe product that still has many characteristics similar the raw unprocessed milk [57,58].

## 3. Animal Welfare and Its Consequences for Legislation

Animal welfare is an increasing issue of public concern and debate. Therefore, many countries are reconsidering their legislation and rules for housing and care of animals. An important factor is that general attitudes about animal welfare are changing. Previously “anthropocentric” thinking was central. Nowadays, there is also place for the insight that “allostasis”, the ability to change, is crucial for good physical and mental health and good animal welfare [59]. As a result, these changing attitudes about animal welfare are accompanied by scientific discussions.

Insufficient awareness of the scientific discussions regarding changing insights in animal welfare may lead to inaccuracies in the legislation concerning animal welfare. Drafting of legislative proposals as to the protection of animal welfare is a complex matter. If the law is not properly substantiated, serious misinterpretations can occur resulting in the exclusion of animals in the country in question. Below we indicate how the dromedary runs the risk of no longer being welcome in Europe on the basis of misinterpreted scientific literature by the legislator in the Netherlands.

## 4. Will the Dromedary No Longer Be Welcome in Europe?

### 4.1. Domestic and Hobby Animals List

On 6 July 2022, the Dutch Minister of Agriculture, Nature and Food Quality published a “positive” list of mammals that can be kept as domestic or hobby animals [60]. Keeping animals that are not on that list will not be allowed anymore from 1 January 2024 onwards. It seems likely that similar legislation will soon apply in other EU countries as well [12]. Animals that are not on the positive list may not be used for production purposes either. Therefore, dromedaries are not only excluded from being kept as domestic or hobby animal, but it would also imply that dromedary dairies will have to stop their activities. This means the end of a thriving European agricultural industry characterized by very low ammonia emission (only about 10% of that of cows [19]) and lower methane emission [23], sustainability [5] and innovation [61].

The purported objective of the list is “to regulate the keeping of animal species as domestic or hobby animals, by prescribing which animal species can be kept as domestic or hobby animals according to the risk of the various animal species on the impairment of animal welfare or of danger to humans or animals” [62].

In 2015, a similar positive list was adopted, but in 2017 the Trade and Industry Appeals Tribunal ruled that the decision-making did not meet the European law requirement of scientific objectivity and that the principles of expertise and transparency had not been respected. Thereupon, in order to establish a new positive list, an independent committee of experts formulated an assessment framework. Based on this framework another independent committee (referred to as the “advisory committee”), assessed 300 different mammalian species and advised which mammals could be placed on the positive list. It is foreseen that the same framework will also be used later to assess birds, reptiles and amphibians [62]. The advisory committee understandably recommended to admit all animal species to the positive list which are at an advanced stage of domestication, even if they are associated with hazards in several risk categories for human or animal health. Non-domesticated or insufficiently domesticated animals were only admitted if they do not represent unacceptable hazards for human and animal health [62]. Several risk categories were defined (Table 2) and it was decided to exclude non domesticated animals that were associated with four or more risk categories according to the decision tree depicted in Figure 2.

According to the committee, the dromedary is not (sufficiently) domesticated and represents a risk in more than three categories. However, the other domestic representative of the genus *Camelus*, the Bactrian camel, was admitted to the positive list. Because the conclusions of the advisory committee seem rather counterintuitive, we decided to conduct a critical review on the biology and domestication of the dromedary camel including the publications that were referred to in the assessment by the advisory committee.

### 4.2. Domestication Status of Camelus dromedarius

Camels have not always been in the desert, On the contrary, their middle and late Miocene ancestors preferred savannah grasslands and were also found in the swampy Gulf Coast forests of Texas and Florida [63]. During the Holocene, the wild ancestors of dromedaries occurred in considerable numbers on the Arabian Peninsula, where they found suitable habitats in coastal regions including mangrove forests [64]. Only after their domestication around 3000–4000 ya in the Southeast coast of the Arabian Peninsula [1], modern domesticated dromedaries were re-introduced into Africa, since the pre-historic giant representatives of *Camelus* had become extinct during the late Pleistocene [65,66,67]. Thus, contrary to the decision of the advisory committee that dromedaries cannot be considered to be sufficiently domesticated yet, the modern species *Camelus dromedarius* is by definition domesticated [68].

Whether initiated by humans or by the animal, intentional or not, the first step in domestication is always docility, the reduced fear of humans [69]. The initial selection of tameness is followed by a suite of morphological and physiological traits that mark the domestic state and are shared amongst many species, but not seen in the wild ancestors: changes in coat color, floppy ears, smaller jaws and teeth (signs of neotenization), modified reproductive cycles, altered hormone and neurotransmitter levels. These changes in behaviour and morphology are collectively referred to as domestication syndrome (DS) or domestication phenotype [70,71,72]. In general, two—not mutually exclusive—hypotheses have been proposed underlying the DS. The neural crest cell (NCC) hypothesis postulates that an initial selection for tameness leads to a reduced function of neural crest-derived tissues relevant for behaviour via mild loss-of-function mutations in neural crest cells (NCCs). Subsequently this neural crest hypofunction produces unselected by-products, such as the morphological changes [70,71]. Genomic support for the NCC hypothesis was found in cats, where genomic regions under selection were associated with NCC survival, neurotransmitters and sensory development [73]. Additionally, during dog domestication, the role of NCC migration and differentiation was highlighted [74]. The thyroid hypothesis (THH) refers to an alteration in the expression of the thyroid hormone triiodothyronine (T3) and its precursor tetraiodo-thyronine (T4), which have key roles in the postnatal and juvenile development and are involved in different pathways, i.a., responsible for growth and maturation [71,75]. The THH has been followed up in chicken [76], cats [73] and dogs [77]. Most importantly, in dromedaries we found genomic signals for both hypotheses [78].

In the study of Fitak et al., 2020 [78] the following sentence in the introduction may have led to a misunderstanding: *“—In essence, domestic Old World camels represent features of the “initial stages” of the domestication process, which were primarily focused on the selection for tameness and docility”*. However, this sentence merely reflects the fact that Old World camels (including dromedary and Bactrian camels) in general have maintained a rather high genetic variability, because the selection pressure to which they were submitted was mainly limited to tameness and docility [79]; whereas many other domesticated animal species were also selected for additional positive traits such as milk or meat production, which led to a secondary bottleneck with concomitant reduced genetic variability.

Another element that contributed to the relatively high genetic variability is the fact that dromedaries were used for transport purposes which facilitated genetic exchange between different populations [80]. Obviously, this does not occur to a large extent with animals that are confined to a much more restricted habitat such as cows, pigs, dogs, etc. From a safety perspective, however, there is no reason to assume that such domesticated animals with a limited gene pool would pose a smaller risk to their environment. What matters is whether traits like tameness and docility are strongly embedded in the genome, which is certainly also the case for the dromedary (as shown in the same publication by Fitak et al., 2020 [78]).

The advisory committee also referred to a publication of Alaskar et al., 2021 [81], however, the scientific relevance in this context is not obvious, except perhaps for a remark that the domestication process of the dromedary is a more recent event. It is true that it started somewhat later than that of the Bactrian camel and horse for instance, but 3000–4000 years of dromedary utilization by humans can be regarded as quite a large time span and more than sufficient to allow for thorough selection of animals with suitable behaviour. Consequently, there is no doubt that the dromedary should be regarded as a domesticated animal. After more than 4000 years its state of domestication is advanced and, for instance, is similar to that of the Bactrian camel, a species that was not excluded from the list.

### 4.3. Risk Assessment

The advisory committee identified several hazards that are associated with keeping dromedaries, but a critical review of the literature revealed serious flaws (Table 2). Of particular importance is the failure to recognize that efficient control measures are available to control the spread of MERS, which implies that it cannot be qualified as a “very high risk zoonotic pathogen”. The physical threat of dromedaries for humans is comparable to that of other large production animals (which is considered to be socially acceptable), and keeping these animals does not pose major concerns for their wellbeing, even under the climatic conditions of The Netherlands.

**Table 2 animals-13-00047-t002:** Summary of comments on the advisory committee’ assessment of risk categories that the dromedary poses for human and animal safety and health.

Risk Category	Advisory Committee’s Assessment [82]	Comment
Danger to humans (zoonoses)	Dromedaries may carry the very high-risk zoonotic pathogen MERS-CoV [83,84] Nearly impossible to prevent spreading of MERS Dromedaries may also carry various other high-risk zoonotic pathogens such as rabies virus [85,86,87], Rift Valley fever virus [88], Brucella abortion, B. meletensis [88], Chlamydia abortion [89], Leptospira interrogans [90] and Mycobacterium bovis [91,92,93]	Correct. However, worldwide prevalence is low and constantly declining [94]. It does not occur in European dromedaries and there is an import ban on dromedaries from outside the EU. At the FAO meeting on “Qualitative risk assessment of MERS-CoV”(Cairo, 5–6 July 2022) the risk of introduction of MERS into Europe was estimated as being virtually nil (B.Faye, unpublished data), notably because the mail live camel flows are regarding the supply for meat market and racing camels from Africa to Arabian Peninsula and not the reverse. Moreover, the only source of live camels for Europe is the Canary Islands (Spain) where MERS-CoV is not present [95].
		Not correct. Efficient prevention of spreading can be achieved by culling, transport restrictions and standard sani-tary measures in hospitals.
		Correct (but the same holds true for other animals on the positive list).
	Control measures are available and applied successfully	Correct
Danger to humans (personal injury)	With dromedaries there is a risk of very serious injury to humans, as a result of which the dromedary falls directly under risk class F ^1^. The dromedary weighs 400–600 kg [96]. During the rutting season, males behave more aggressively, and can attack humans by biting, for example. This can lead to fatal injuries [96,97]. Given the size and behaviour of dromedaries, they can cause very serious injury to humans, placing the dromedary directly under risk class F ^1^.	Not correct. The risk of dromedaries causing personal injury to humans [98] is comparable to that of other large, domesticated animals such as horses and cows. It should be noted, though, that the domestication process has contributed to reducing the probability of the occurrence of serious events to what is considered as socially acceptable (i.e., similar to the probability of being injured by horses, cows or camels)
Food intake ^2^	The dromedary has hypsodont molars [99,100]. Therefore, this risk factor is present. Dromedaries forage 8–12 h a day and spend rest periods ruminating [100]. Dromedaries live in (semi)arid areas [98], where food and water are limited [101,102], which means that they cover great distances, up to about 50–70 km per day [103]. Therefore, this risk factor applies.	Correct. However, this is not a problem with correctly balanced feed [104]. Furthermore dromedaries stand and grind their teeth for 8–12 h a day [100]. This prevents hypsodonty, the lifelong growth of teeth and molars, as occurs for example in rabbits [105].
		Not correct Although dromedaries forage for 8–12 h a day, they do not have to travel long distances for water and food in Europe. Of course excess food should be prevented
Thermoregulation ^2^	Dromedaries live in a dry tropical and subtropical climate [97,106]. In the dry tropical and subtropical climate, with few regional exceptions, the average monthly temperature is above 10 °C throughout the year. In some areas, the average monthly temperature of the coldest month falls to 5 °C. During 5–12 months of the year, the average temperature is above 18 °C. The average annual precipitation varies, but is up to 500 mm. The dromedary is very sensitive to humidity [97], which prevents pneumonia [107]. The dromedary is adapted to a dry tropical and subtropical climate. This risk factor therefore applies.	Not correct. Dromedaries are known for their very strong adaptation ability and remarkably good thermoregulation [108]. Some authors have suggested that dromedaries are sensitive to humidity, which would predispose them to pneumonia [100], However, there is no evidence that this would hold true for dromedaries in Europe. Furthermore, not humidity, but lack of good housing allowing good protection of the camel is the main risk factor. Pneumonia in dromedaries is no more common than in other large farming animals such as horses and cows according to the veterinarians Peter Klaver and Roland van Riel (unpublished data).
Social behaviour ^2^	Dromedaries live in herds of only females, only males, a mixed population or solitary. In the most common structure, herds with one male and several females, the male leads the herd and guards the females against competing males. There is a despotic dominance hierarchy [100,109]. This risk factor therefore applies.	Not correct. The despotic dominance hierarchy present in dromedaries [100] plays a favorable role, except when several male dromedaries are in the vicinity of adult female dromedaries during the rutting period, which can be prevented by appropriate risk management [110].

^1^ Keeping specimens of animal species in risk class F poses a high risk to people’s health. ^2^ These risk categories affect animal welfare.

### 4.4. Appeal to the Ministry’s Decision to Ban Dromedaries from the “Positive List” and Follow Up

In discussions with the responsible officials of the ministry it was emphasized that the literature referring to the domestication status and the risk of MERS spread might have been misinterpreted. Furthermore, concerning the assessment of the risk categories for animal health, the specific properties of the dromedaries were not sufficiently taken into account. Consequently, the dromedary should be reconsidered to be included in the domestic and hobby animals list.

The committee replied to the ministry that they deleted the characteristic *that an animal species is domesticated when the non-domesticated species is extinct*, from the definition of domestication of Nijenhuis and Hopster, 2018 [68]. They did so because there are species of which the non-domesticated animal became extinct, that could not be admitted to the “positive list”, such as the Przewalski’s horse. In this context, we would like to draw attention to the among experts widely accepted definition of domestication by Zeder et al., 2015 [111] stating *”Domestication is a sustained multigenerational, mutualistic relationship in which one organism assumes a significant degree of influence over the reproduction and care of another organism in order to secure a more predictable supply of a resource of interest, and through which the partner organism gains advantage over individuals that remain outside this relationship, thereby benefitting and often increasing the fitness of both the domesticator and the target domesticate”*. This definition clearly recognizes the dromedary as domesticated. Intriguingly, the advisory committee argued that dromedaries would be rather habituated than domesticated, which should explain the ease with which they survive in the wild. However, this opinion was not substantiated with scientific publications. Therefore, a PubMed search was carried out, using “dromedary” and domestication which yielded 527 publications. Using “dromedary” and “habituation”, we did not find any scientific evidence that the dromedary can be considered as rather habituated than domesticated. Finally, the genomic selection signals found in dromedaries cannot be explained by habituation but rather by adaptation [112] and long term domestication [78].

Does reliable sources (=primary, peer-reviewed scientific literature) show that specimens of the species in question are kept by humans for many generations?

In the circumstances described, is there targeted, consistent selection and intensive breeding of individuals with human-useful characteristics and traits?

Has this breeding over generations caused stable changes in behaviour and/or morphology and/or physiology and/or reproduction in the animal species or population concerned, with which it demonstrably distinguishes itself from the original wild type?

All these questions can be answered with “yes” and thus further confirm the advanced domesticated stage of the dromedary.

## 5. Discussion

The above-described increase of the camel milk market is in accordance with the transition from the multi-purpose dromedary in nomadic environments to a single purpose dromedary in specialized dairies [5] and the advancements in camel milk processing [8]. The intensification of camel farming systems has in turn stimulated the development of protocols assessing camels’ welfare. These protocols call for thorough evaluation and validation before they will be generally applicable to control camels’ welfare in practice. The same applies to pedigree and genomic databases as currently developed in Europe [42] for preventing progressive impoverishment of the gene pool.

For successful camel milk processing, food safety of camel milk is crucial. Properly applied principles of good manufacturing practices (GMP), hazards analysis critical control points (HACCP) and biosecurity will warrant that dromedary milk hygiene and safety [113] are comparable with those of cow’s milk. It will be necessary to investigate if frequent controls of microbial safety of the produced dromedary milk will make it possible that raw milk can be safely sold unprocessed/unpasteurized. It should be noted that hazards generally associated with raw milk are smaller with camel milk than with cow’s milk. These hazards are mostly due to fecal contamination of udders with enteric bacteria [113]. The chance that camel feces will contaminate udders is smaller than that of cow’s feces, because the consistence of small, dry and firm camel feces balls, differs considerably from the big, wet cow’ feces. Moreover, the implantation of udder is less pendulous than in dairy cow and thanks to its sternal pad, the udder is less in contact with the soil when the camel is sitting. The availability of raw camel milk would meet the consumers’ demand for completely unprocessed milk. Preference of raw milk does not only originate from traditional users, but also from consumers who prefer raw milk because of the demonstrated property of cow’s milk to reduce allergy and asthma [114]. The farmers must always be aware of their responsibility for the safety of the product, while public health authorities have a role in protecting the consumers by ensuring that the farmers take this responsibility seriously.

Camel milk sale is expected to increase considerably if scientific research should confirm one or more of its hypothesized health benefits.

Dromedary dairies do not have to be large-scale. Milking dromedaries can be combined well with other activities in small-scale settings. In Europe, multifunctional agriculture is receiving a great deal of attention nowadays. Milking of dromedaries is then combined with touristic activities (agritourism) or with care of children or people with a physical or mental disability. Supervised tourist/guests can carry out simple animal related activities at the farm, or other activities, e.g., bottling milk or preparing camel milk products, such as dromedary milk ice cream.

Dromedaries are recognized as one of the most sustainable livestock animals [5]. Their sustainability, which is already high due to their low ammonia emissions [19] and less methane emission [23], may increase further in case scientific research can confirm the health promoting effects attributed to dromedary urine. It will then be economically attractive to collect urine from the dromedaries before it comes into contact with feces. Beyond, this process will contribute to an overall decreasing ammonia emission. Sustainability can also increase when dromedary food intake is protein reduced, because excess nitrogen in proteins waisted and unnecessarily increases ammonia emission [115].

The increased camel milk market and the intensification process of camel farming systems stimulates the ongoing development of legislation controlling welfare of camels. This legislation needs to be based on the scientific knowledge of behavioral, physiological and genetic properties of camels [116]. However, scientific evidence can be easily misconceived, as happened to the Dutch government. In contrast to what is suggested by the Dutch advisory committee there is ample evidence that the dromedary is at an advanced stage of domestication [78,111]. Experiences of dromedary dairy farms in Europe have shown that these animals can be managed adequately without significant risks for humans, the environment and the animals themselves. This calls for a timely consultation of camel experts. Finally, legislation for animal welfare is still in its infancy and is therefore prone to misinterpretations.

Mutual awareness of both breeders and legislators for the rapidly increasing knowledge about dromedaries and their products strengthen the position of camels as one of the most adapted and sustainable livestock animals.

## 6. Conclusions

Camel dairies in Europe produce camel milk that complies with the same food safety rules as for cow’ milk. Recently developed protocols can control camels’ welfare after thorough evaluation and validation. Legislation to guard welfare of camels is still in its infancy and needs to be based on cutting-edge scientific knowledge about camels. Timely consultation of camel experts in a culture in which evidence-based policy and decision making is good practice, will help legislators to safeguard welfare of camels. The presumed beneficial properties of camel milk as “superfood” as well as scientific studies, which evaluate its basis and identify genomic regions underlying milk production traits, will support the young camel milk market in Europe to flourish.

## Figures and Tables

**Figure 1 animals-13-00047-f001:**
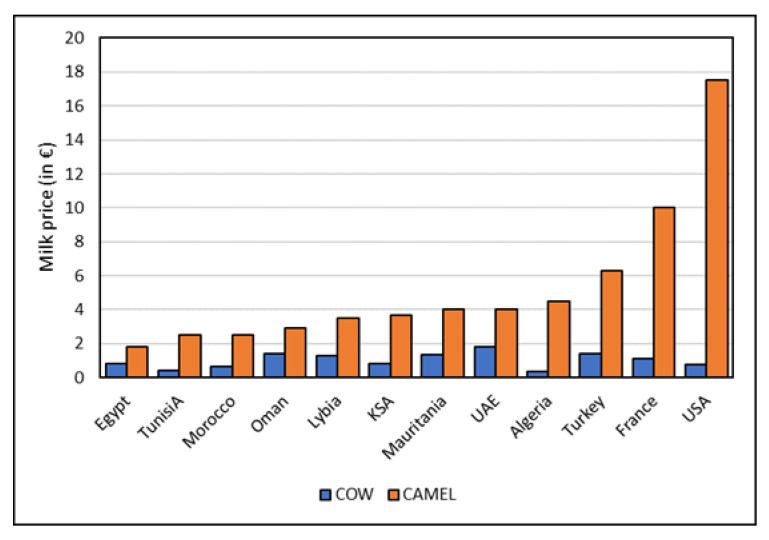
Differential prices of dromedary camel and cow milk in different countries.

**Figure 2 animals-13-00047-f002:**
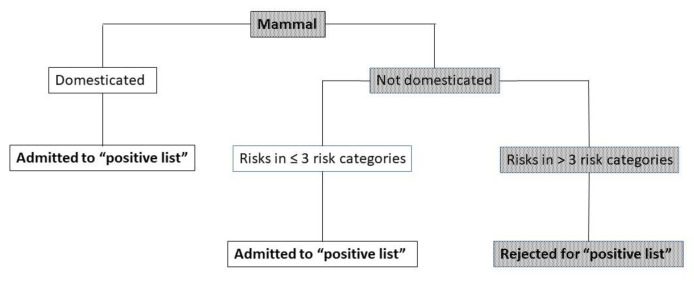
Decision tree for the admittance to the domestic and hobby animals list (“positive list”. The arched text represents the advisory committee’s assessment of the dromedary.

**Table 1 animals-13-00047-t001:** Presumed medicinal properties of camel milk, ascribed working mechanisms and type of most relevant studies, based on PubMed search using the words camel milk and the presumed medicinal property.

Health Problem	Presumed Beneficial Mechanism of Camel Milk	Type of Study	Reference
Diabetes	Influence at immune system Insulin-like protein in milk	Randomized cow’s milk controlled	[30,31]
Cow’s milk allergy	Absence of betalactoglobuline in milk; other structure of lactose.	Cross-over study Observational study	[32,33]
Cancer	Antioxidant properties of milk	Observational	[34]
Hepatitis C	Anti-viral and immunomodulatory activities Immunoglobulins and lactoferrin	Observational	[35]
COVID-19	Lactoferrin	Hypothesis	[36]
Herpes simplex	Lactoperoxidase	In Vitro	[37]
Gastro-intestinal disorders	Lactoferrin and immunoglobulins	Hypothesis	[38]
Autism	Vitamins, minerals and immunoglobulins	Meta-analysis of randomized controlled trials ^1^	[39]

^1^ Very short treatment time frames, not extending more than 2 weeks

## Data Availability

Not applicable.

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
