# Peer review of "The Flourishing Camel Milk Market and Concerns about Animal Welfare and Legislation"

_animals, 2022, doi:10.3390/ani13010047_

Round 1
Reviewer 1 Report
This article perfectly describes the strengths, weaknesses, opportunities, and threats (like a SWOT analysis) of increasing camel milk production worldwide. This conversion from a multi-purpose specie to a dairy farming specie brings new challenges and possibilities that the article explains very well. In addition, it makes a perfect description of the current situation of recently established camel dairy farms in Europe. Definitely, it is interesting and valuable for developing European countries' new legislation, where the introduction of this specie (for dairy farming) is recent.
Comment and Suggestions for Authors:
It is an interesting article that addresses the current context at a European and worldwide level, in which camel milk production has increased by more than 8% per year. The current situation means new challenges and, at the same time, new opportunities for camel milk.
Specific comments:
The main question addressed in this article (explained before) is very well defined in the title and the references used are appropriate to the arguments exposed.
The selected topic defines the current situation of the camel milk sector and the problems it faces. The article also proposes solid scientific arguments which could contribute to changing the legislation of some European countries that negatively affect this sector. It could be helpful to the competent authority in charge of preparing the regulations.
The conclusions perfectly reflect the arguments presented. The interest of this paper is clear.
There are no specific comments for improvement or clarifying aspects of this article.
Author Response
Point: There are no specific comments for improvement or clarifying aspects of this article.
Response: We thank the reviewer for the favorable comments
Reviewer 2 Report
The current paper entitled “The flourishing camel milk market and its consequences for animal welfare and legislation”is fairly uncommon presentation. The manuscript relates two main concerns, the growing European dromedary milk market and the Dutch legislator exclusion of camel from the “positive list”. The two queries seem to be connected; but, somehow, controversial. Moreover, it appears clearly that the promising camel milk market is not a consequence for the abused Dutch legislation. Therefore, the title does not reflect perfectly the content of the current paper.
The introduction described the back history of the genus camelus, the process of its domestication and its utilization particularly as animal milk producer. It also evoked the adoption of the “positive” domestic and hobby animals’ lists.
The second part was dedicated to the evolvement and promising of camel milk market within the European countries. This part was quite exhaustive and unnecessarily included some data on camel physiological adaptation.
The third and forth parts were devoted to the welfare of dromedary camel and the relevant legislation. The authors attempted to justify the domestic status of this species and to emerge different criteria backing the admission of dromedary camel within the domesticated and hobby animals list. The presented arguments were fully persuasive that the Dutch legislator should reconsider the decision of excluding the dromedary camel from the “positive list”.
The discussion part was mainly devoted to camel milk production and market; while, only a small paragraph was reserved to the legislation matter.
As a whole, the manuscript is well written. Some minor rectifications are included in the attached version of the manuscript. In my opinion, the arguments submitted concerning the status of this species are convincing and concluding.

Author Response
We thank the reviewer for the clear comments.
Point 1:
The current paper entitled “The flourishing camel milk market and its consequences for animal welfare and legislation”is fairly uncommon presentation. The manuscript relates two main concerns, the growing European dromedary milk market and the Dutch legislator exclusion of camel from the “positive list”. The two queries seem to be connected; but, somehow, controversial. Moreover, it appears clearly that the promising camel milk market is not a consequence for the abused Dutch legislation. Therefore, the title does not reflect perfectly the content of the current paper.
Response 1: In fact the title wants to say that the article describes the growing camel milk market and that this growing camel milk market has significant influence on animal welfare and legislation. Indeed the title can be improved. Therefore, we changed the title into: "The flourishing camel milk market and concerns about animal welfare and legislation”
Point 2: The second part was dedicated to the evolvement and promising of camel milk market within the European countries. This part was quite exhaustive and unnecessarily included some data on camel physiological adaptation.
Response 2:
We included data on camel physiological adaptation to explain specific problems with breeding and milking, because they explain the high price of camel milk.
point 3: The discussion part was mainly devoted to camel milk production and market; while, only a small paragraph was reserved to the legislation matter.
Response 3: We tryed to comment the legislation matter as concise as possible.
Point 4: As a whole, the manuscript is well written. Some minor rectifications are included in the attached version of the manuscript. In my opinion, the arguments submitted concerning the status of this species are convincing and concluding.
Response 4:
The reviewer 2 has proposed to change the way how names are written: instead of “Camelus dromedaries” (s)he proposes “camelus dromedaries”. This is not in line with what is recommended by the International Commission on Zoological Nomenclature (ICZN): https://www.iczn.org/outreach/guidelines-for-authors-and-editors/whats-in-a-name/ so I think this does not have to be adapted. Besides, when I checked this in a recent article in “ Animals” it appears that this journal also adheres to the ICZN recommendations.
The other minor rectificatios are included in the manuscript.
Reviewer 3 Report
This paper by Marcel et al.investicated the camel milk market, and summarized the animal welfare and legislation. In recent years, there are more and more researches on camel milk. However, research on camel milk market is rare. In this paper, auther gives the reasons for the increase of the camel milk market, animal welfare and legislation. This paper corresponds to the current trend of camel milk market development, and has certain reference value. The content layout is reasonable and well written.
Author Response
Point:
This paper by Marcel et al.investicated the camel milk market, and summarized the animal welfare and legislation. In recent years, there are more and more researches on camel milk. However, research on camel milk market is rare. In this paper, auther gives the reasons for the increase of the camel milk market, animal welfare and legislation. This paper corresponds to the current trend of camel milk market development, and has certain reference value. The content layout is reasonable and well written.
Response: We thank the reviewer for this favorable comment